# De Novo Generation of Human Hematopoietic Stem Cells from Pluripotent Stem Cells for Cellular Therapy

**DOI:** 10.3390/cells12020321

**Published:** 2023-01-14

**Authors:** Jianyi Ding, Yongqin Li, Andre Larochelle

**Affiliations:** Cellular and Molecular Therapeutics Branch, National Heart Lung, and Blood Institute (NHLBI), National Institutes of Health (NIH), Bethesda, MD 20892, USA

**Keywords:** hematopoietic stem cell, pluripotent stem cell, embryonic stem cell, reprogramming, differentiation, intrinsic cues, extrinsic cues, cellular therapy, development, cell signaling, transcription factor

## Abstract

The ability to manufacture human hematopoietic stem cells (HSCs) in the laboratory holds enormous promise for cellular therapy of human blood diseases. Several differentiation protocols have been developed to facilitate the emergence of HSCs from human pluripotent stem cells (PSCs). Most approaches employ a stepwise addition of cytokines and morphogens to recapitulate the natural developmental process. However, these protocols globally lack clinical relevance and uniformly induce PSCs to produce hematopoietic progenitors with embryonic features and limited engraftment and differentiation capabilities. This review examines how key intrinsic cues and extrinsic environmental inputs have been integrated within human PSC differentiation protocols to enhance the emergence of definitive hematopoiesis and how advances in genomics set the stage for imminent breakthroughs in this field.

## 1. Hematopoietic Stem Cells for Cellular Therapy

### 1.1. Limited Donor Hematopoietic Stem Cell Availability for Cellular Therapy

Hematopoietic stem cells (HSCs) are defined by their remarkable ability to reconstitute and maintain a functional hematopoietic system over extended periods of time following transplantation into conditioned hosts. This endowment of HSCs to sustain long-term hematopoiesis relates to their ability to either differentiate to produce mature progeny of all myeloid and lymphoid blood cell lineages or self-renew to replace cells that become progressively committed to differentiation. HSCs are thus important cellular targets for the permanent correction of inherited hematologic, metabolic and immunologic disorders. However, allogeneic HSC transplant options are often impeded due to insufficient numbers of transplantable HSCs in donor umbilical cord blood (UCB) units or mobilized peripheral blood (MPB) cell collections, as well as the limited availability of human leukocyte antigen (HLA)-matched donors. Sufficient autologous HSCs for genetic modification are also commonly unavailable in patients with impaired hematopoiesis, such as marrow failure syndromes and chronic inflammatory disorders (Figure 1). Hence, strategies to circumvent these shortcomings would have significant clinical implications.

### 1.2. Approaches to Increase HSC Availability

Two general tactics are in development to enhance available sources of HSCs for therapy (Figure 1). Ex vivo expansion of HSCs with long-term repopulating potential could provide a clinically valuable strategy to increase cell doses and therapeutic efficacy. However, safe, effective and clinically feasible expansion methodologies remain unavailable. The current state of ex vivo HSC expansion for clinical applications has been reviewed elsewhere [1,2,3]. De novo generation of HSCs provides an alternative tool for the development of cell-based therapies, and offers additional applications for disease modeling and drug testing when primary cells from patients are limited [4]. The process requires manipulation of cell fate to directly or indirectly reprogram adult somatic cells to an HSC phenotype.

**Figure 1 cells-12-00321-f001:**
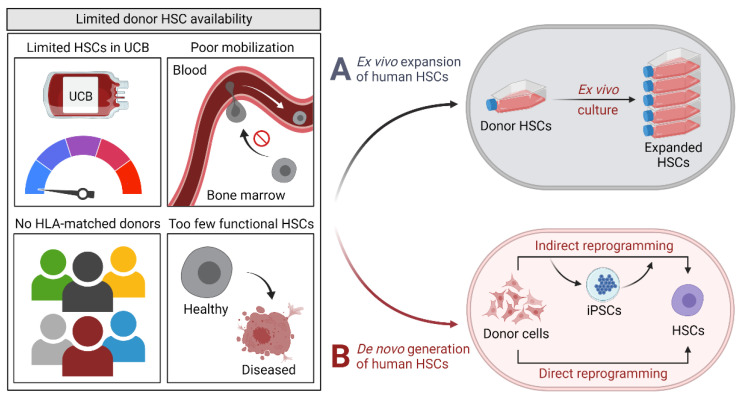
**General approaches to increase donor hematopoietic stem cell availability for cellular therapy**. Limited donor hematopoietic stem cell accessibility (e.g., due to insufficient numbers in donor umbilical cord blood units or mobilized peripheral blood cell collections, lack of human leukocyte antigen-matched donors, and poor functionality in patients with impaired hematopoiesis) may be overcome by ex vivo expansion or de novo generation by direct or indirect reprogramming of adult somatic cells. Abbreviations: UCB: umbilical cord blood; HSCs: hematopoietic stem cells; iPSCs: induced pluripotent stem cells; HLA: human leukocyte antigen.

#### 1.2.1. Indirect Reprogramming

An indirect reprogramming approach that relies on a pluripotent state to change cellular fate was conceptualized in 2006 with Yamanaka’s landmark induction of pluripotency in murine fibroblasts by a defined set of transcription factors (Figure 1) [5,6,7]. This work provided unambiguous evidence that the epigenetic state of differentiated cells can be reversed; the cells generated through this process were termed induced pluripotent stem cells (iPSCs). Investigators soon applied similar methodologies to generate human iPSCs with pluripotent properties comparable to native human embryonic stem cells (ESCs) [8,9]. Because human iPSCs are engineered ex vivo from somatic cells, they avoid the ethical implications of harvesting human embryos for ESC derivation from the inner cell mass of developing blastocysts. In addition, with the inception of human iPSCs came the appealing concept of generating human iPSCs from an individual patient, correcting the genetic defect and differentiating the disease-free iPSCs into a theoretically infinite supply of transplantable autologous HSCs. The proof-of-principle of using iPSC technologies to cure hematopoietic disorders attributed to a genetic defect has been performed in animal models [10,11,12,13,14], but obstacles still hinder clinical translation of this approach.

Several basic protocols have been developed to promote hematopoietic differentiation from human pluripotent stem cells. In one approach, iPSCs or ESCs are cultured directly on supportive stromal layers (see Section 4.1). Prolonged culture of these pluripotent stem cells (PSCs) on stromal cells resulted in the formation of inflated sac-like structures containing hemangioblast-like progenitors with differentiation potential toward platelet and erythroid fates [15,16,17]. However, the animal origin and tumorigenic potential of the immortalized stromal cells used in these studies represent intractable drawbacks for the clinical translation of this approach. An alternative strategy utilizes extracellular matrix proteins to generate a culture system that supports the generation of hematopoietic cells from human pluripotent stem cells (see Section 4.2). This system offers advantages of simplicity, higher consistency and safety, but it is unlikely sufficient to recapitulate the environment necessary for hematopoietic development. Another commonly used differentiation platform relies on the formation of three-dimensional aggregates, known as embryoid bodies (EB), in suspension cultures of pluripotent stem cells [18]. This method simulates early stages of post-implantation embryonic development by promoting the spontaneous generation of all three primordial germ layers within each aggregate. These tissue types enhance hematopoietic development by providing key microenvironmental signals to differentiating PSCs. However, differentiation outcomes are highly variable and dependent upon the size and quality of EBs. In addition, hematopoietic cell yield is generally modest, and subsequent plating of EB-differentiated cells on gelatin [19] or OP9 cells [20] is often required to promote maturation and boost hematopoietic cell production. Recently, monolayer-based differentiation protocols have also become available [21,22,23]. During differentiation, a supportive PSC-derived adherent monolayer rapidly forms, followed by the emergence of suspension hematopoietic cells that can be harvested at regular intervals during culture for analysis. The utility of this approach hinges on its scalability, adaptable experimental design and inherent simplicity, requiring no re-plating, EB formation or co-culture on stromal elements. Monolayer differentiation systems reproducibly yield enriched populations of hematopoietic progenitors, but further optimization is required to promote differentiation of PSCs into fully functional HSCs.

#### 1.2.2. Direct Reprogramming

The terms direct reprogramming, transdifferentiation or conversion have been coined to describe the process of induction of a desired cell lineage from adult somatic cells without transitioning through an intermediary iPSC state (Figure 1). The first evidence of direct conversion of cellular identity was provided in the late 1980s by transient overexpression of the skeletal muscle transcription factor MyoD to induce in murine fibroblasts phenotypic attributes normally found within the myoblastic lineage [24]. Since the initial publication of this report, additional examples of direct cellular reprogramming have emerged (reviewed in [25]).

#### 1.2.3. Pros and Cons of Reprogramming Approaches

While direct approaches bypass the complexity of iPSC generation and could, in theory, enable reprogramming of cells in vivo, the efficiency of conversion remains low, safe delivery methods for in vivo transdifferentiation are nonexistent, the limited expansion potential of the post-mitotic starting cell populations may preclude large scale production ex vivo, and the in vitro differentiated cells largely retain an immature phenotype. The indirect approach presents similar pitfalls with modest efficiency of reprogramming and immaturity of the differentiated cellular populations. However, iPSCs can be expanded indefinitely in culture for large-scale clinical production and are readily amenable to genetic manipulation utilizing integrating viral vectors or programmable engineered nuclease technologies. Moreover, the development of a universal, immune-compatible, off-the-shelf product could reduce the inherent costs and slow turnaround time associated with individualized iPSC production, and thus maximize the likelihood of translation to a clinical setting. This review provides a synopsis of recent advances in the indirect reprogramming of human adult somatic cells toward a mesodermal hematopoietic lineage, with an emphasis on therapeutic applications for inherited disorders of the blood system.

### 1.3. Readout Assays to Assess the Generation of Functional HSCs in PSC Cultures

Transplantability is the conventional criterium to confirm HSC identity after PSC differentiation. To measure human HSC activity, long-term, multilineage engraftment after primary and secondary transplantation into xenograft immune-deficient mouse models represents the current gold standard and is the most stringent assay available. However, the costly and time-consuming in vivo transplantation method is cumbersome and precludes rapid high-throughput assessment of functional HSCs in PSC differentiation cultures. Lymphoid differentiation potential of PSC-derived cells is commonly viewed as an exclusive attribute of HSCs and used as a surrogate for HSC generation ex vivo. However, lympho-myeloid biased progenitors without engraftment potential are known to arise before HSCs in vivo [26,27,28,29,30]. Thus, lymphoid potential alone is insufficient to predict the presence of engraftable HSCs in culture. Demonstration of embryonic or fetal globin switch to adult β-globin protein in erythroid cells differentiated from PSCs through an intermediate hematopoietic progenitor provides evidence of increased maturity of the cellular product but is inadequate to identify transplantable HSCs. Detection of CD34+CD38-CD45RA-CD90+ cells by flow cytometry in human PSC cultures is also customarily used to confirm the generation of a cell population highly enriched in human HSCs. This phenotypic definition offers a useful tool to characterize hematopoietic differentiation in human PSC cultures but generally correlates poorly with HSC function. Other in vitro systems, including colony-forming unit (CFU) and long-term bone marrow culture (LTBMC) assays, are routinely employed to assess efficacy of hematopoietic cell formation in PSC differentiation protocols. However, these approaches detect committed progenitors with biological properties distinct from multipotent HSCs. Integration of single-cell gene expression profiles of hematopoietic cells generated ex vivo with the recently established single-cell transcriptome landscape of human HSCs during ontogeny [30] is poised to provide a more practical approach to identify functional HSCs in human PSC differentiation cultures.

## 2. The Hematopoietic Developmental Process

Approaches to derive HSCs from induced or embryonic pluripotent stem cells in vitro generally attempt to recapitulate the developmental process that occurs during embryonic and fetal development [31]. Advances in various vertebrate models, including frog, quail, chick, zebrafish and mouse systems, have led to key fundamental knowledge on the stepwise processes of hematopoiesis and the complex array of molecular, cellular and local developmental cues that regulate hematopoietic development. While the direct extrapolation of findings derived from model organisms is not always fitting owing to species differences, the sophisticated understanding gained from these studies has provided highly valuable information for the generation of human HSCs from PSCs ex vivo. Additionally, novel single-cell genomic technologies are being used to gain unique insights into developmental hematopoiesis and identify missing intrinsic and extrinsic cues required for HSC generation ex vivo [30,32,33,34,35,36,37,38,39]. This section highlights how these approaches have contributed to our understanding of major developmental steps and regulatory signals implicated in specification of hematopoiesis from ESCs through stages of mesodermal induction, hematoendothelial specification, vascular arterialization, endothelial-to-hematopoietic transition and maturation (Figure 2).

### 2.1. Mesodermal Induction

The process of blood formation during embryonic mammalian development begins with a series of commitment steps within the inner cell mass of the blastocyst. The hypoblast and epiblast layers that form at this stage will later give rise to the yolk sac (YS) and the embryo proper, respectively. A morphological groove termed the primitive streak then appears on the posterior epiblast, marking the commencement of gastrulation [40,41,42,43].

The process of gastrulation results in the formation of a trilaminar germ disc, consisting of the ectoderm, mesoderm and endoderm primordial germ layers. In vivo, the circulatory system (heart, blood and vascular endothelium) derives from mesodermal cells found at the periphery of the embryo within the splanchnic lateral plate mesoderm. As such, the process of mesodermal induction marks the onset of hematopoiesis in the embryo. Various developmental biology approaches have identified key regulatory networks implicated in this process. Most notably, BMP, FGF, Activin/Nodal and canonical WNT intracellular signaling pathways have distinct and overlapping roles in the induction of mesoderm during development [7,44] (Figure 2).

**Figure 2 cells-12-00321-f002:**
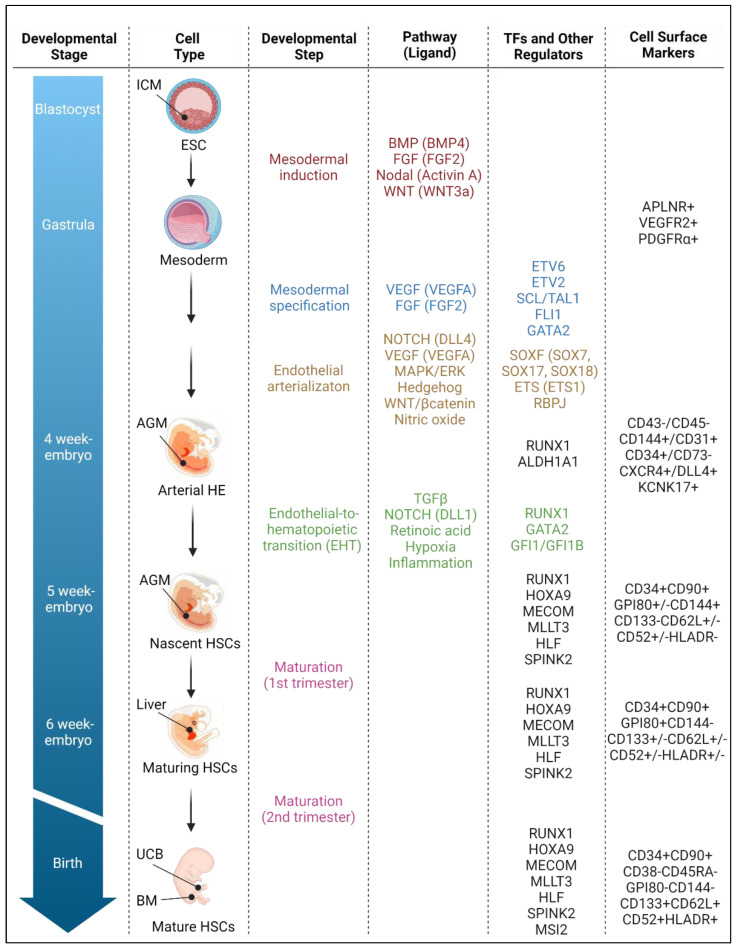
**Human hematopoietic developmental process**. Specification of hematopoiesis from ESCs occurs through stages of mesodermal induction, hematoendothelial specification, vascular arterialization, endothelial-to-hematopoietic transition and HSC maturation during development. Key intrinsic regulatory networks and cell surface epitopes mark each developmental step in vivo. Most approaches for de novo generation of HSCs from PSCs attempt to recapitulate in vitro the process that occurs during embryonic and fetal development. Abbreviations: AGM: aorta–gonad–mesonephros; ALDH1A1: aldehyde dehydrogenase 1 family member A1; APLNR: apelin receptor; BM: bone marrow; BMP: bone morphogenetic protein; CD: cluster of differentiation; CXCR4: C-X-C chemokine receptor type 4; DLL4: delta-like 4; EHT: endothelial-to-hematopoietic transition; ESC: embryonic stem cell; ETS: E26 transformation-specific; ETV: translocation-ETS-leukemia virus; FGF: fibroblast growth factor; FLI1: friend leukemia integration 1 transcription factor; GATA2: GATA binding factor 2; GFI1: growth factor independent 1 transcriptional repressor; GPI80: glycosylphosphatidylinositol-80; HE: hemogenic endothelium; HLA-DR: human leukocyte antigen-DR; HLF: hepatic leukemia factor; HOXA9: homeobox protein A9; HSCs: hematopoietic stem cells; ICM: inner cell mass; KCNK17: potassium two pore domain channel subfamily K member 17; MAPK/ERK: mitogen-activated protein kinases/extracellular signal-regulated kinases; MECOM: MDS1 and EVI1 complex locus; MLLT3L: mixed-lineage leukemia translocated to 3L; MSI2: Musashi RNA-binding protein 2; PDGFRα: platelet-derived growth factor receptor alpha; RBPJ: recombination signal binding protein for immunoglobulin kappa J region; RUNX1: runt-related transcription factor 1; SCL/TAL1: stem cell leukemia/T-cell acute leukemia protein 1; SOXF: Sry-related high-mobility-group box F; SPINK2: serine peptidase inhibitor kazal type 2; TFs: transcription factors; TGFβ: transforming growth factor-β; UCB: umbilical cord blood; VEGF: vascular endothelial growth factor; WNT: wingless-related integration site.

### 2.2. Mesodermal Specification to Hematoendothelial Fates

Soon after gastrulation, as the developing embryo expands in size and complexity, systemic demands for nutrients, oxygen and waste disposal also evolve. To support the growing fetus, mesodermal progenitors specify hematoendothelial fates, establishing a series of temporally and spatially distinct blood systems designated as “hematopoietic waves”. These successive hematopoietic events first occur in the YS and later within the aorta–gonad–mesonephros (AGM) region of the developing conceptus [45].

#### 2.2.1. Yolk Sac Hematopoiesis

The first wave of hematopoiesis, designated as “primitive”, originates in the early YS blood islands. It is marked by the rapid production of large, nucleated red blood cells that arise from mesodermal precursors and can be distinguished by the expression of primitive embryonic globin proteins. This wave also produces unipotent megakaryocyte and macrophage progenitors [46,47]. The second wave of hematopoiesis also arises within the YS of the developing embryo but supplies progenitors with less restricted erythro-myeloid and lympho-myeloid multilineage differentiation potential [48]. Wave two is termed “definitive” to highlight the persistence of erythroid cells produced at this stage through fetal/postnatal life and the distinct expression of fetal and adult globin proteins in these cells. However, hematopoietic progenitors at this developmental stage lack the long-term engrafting capacity associated with HSCs.

#### 2.2.2. AGM Hematopoiesis

The third wave of hematopoiesis is also labeled “definitive”, but specifically occurs in the embryo proper. It marks the first appearance of multilineage engrafting HSCs primarily within the ventral wall of the dorsal aorta (DA) in the AGM region of the embryo [49,50,51,52]. A close developmental relationship exists between endothelial and hematopoietic cells. Within the AGM, HSCs arise from hemogenic endothelium (HE), a unique subset of unipotent vascular endothelium (VE) fated to a blood lineage identity [53,54,55]. Identification of CD43 as an early marker of hematopoietic cells distinct from the panleukocyte marker CD45 expressed at later stages of development defined an important milestone in the characterization of HE in culture [56,57]. The marker combination enabled precise separation of CD43+CD45+ hematopoietic cells from CD43-CD45- HE. Populations of HE also express well-known endothelial surface proteins, including CD144 (alternatively termed VE-Cadherin, VE-Cad), CD31 (Pecam1) and CD34. Recent functional studies of the CD43-CD45-CD144+CD34+ population in human PSC cultures further demonstrated the distinct lack of CD73 expression in hemogenic endothelium, delineating a CD43-CD45-CD144+CD34+CD73- phenotype to identify HE cells ex vivo [58,59]. In murine [36] and human embryonic vasculature [34,35], CD44, a cell surface glycoprotein that interacts with the extracellular matrix molecule hyaluronan, was also shown to enable more precise isolation and enrichment of HE fated to definitive HSC generation.

#### 2.2.3. Regulatory Determinants of Mesodermal Specification

Vascular endothelial growth factor-A (VEGFA) was identified as a critical initiator signal for the progression of mesoderm to hematoendothelial programs [60]. Transcription factor ETV6 (a member of the ETS family) drives VEGFA signaling [61], and hematopoietic cytokine FGF2 enhances the sensitivity of mesoderm cells to VEGFA by upregulating the surface expression of VEGF receptor-2 (VEGFR-2, alternatively termed KDR or FLK1) on the subset of mesodermal precursors with hematopoietic potential. This endogenous VEGFA signaling synergizes with a parallel pathway controlled at its apex by the ETS transcription factor ETV2 in mammals. Immediately downstream of ETV2, the basic helix–loop–helix factor SCL/TAL1 is indispensable for blood cell emergence through active suppression of alternative fates from mesodermal precursors [62]. Cellular activities of SCL are mediated through nucleation of a core quaternary protein complex (SCL: Protein E: LMO2: LDB1), in an interconnected regulatory loop with other transcription factors of the ETS (FLI1) and GATA (GATA2) families [63,64,65,66,67] (Figure 2).

### 2.3. Vascular Endothelium Arterialization

Hemogenic endothelial cells giving rise to long-term engraftable HSCs have not been associated with venous endothelium during development, suggesting that arterial specification of HE is required for definitive HSC emergence in the AGM [68,69,70]. A direct developmental trajectory from arterial endothelium to HSCs was demonstrated in an scRNA-seq map of human HSC ontogeny [30]. NOTCH signaling is recognized as the most critical determinant of arterial specification in embryonic vasculature [71,72,73]. Arterialization of vascular endothelium is initiated by expression of the NOTCH ligand DLL4 in vertebrate embryos [74]. Patterns of DLL4 expression are regulated by SOXF (e.g., SOX7, SOX17, SOX18), ETS (e.g., ETS1) and RBPJ transcription factor gene families [75,76,77]. The intersection of multiple additional signaling pathways, including VEGF, MAPK/ERK, Hedgehog, WNT/β-catenin and blood flow-induced nitric oxide (NO) [78,79,80], are also implicated in the control of arterial fate specification at this stage of development (reviewed in [81,82]) (Figure 2).

### 2.4. Endothelial-to-Hematopoietic Transition (EHT)

#### 2.4.1. Hematopoietic Transition from HE

The mechanism by which HE cells gain hematopoietic potential and morphology at the expense of their endothelial identity during YS and AGM definitive hematopoiesis is known as endothelial-to-hematopoietic transition (EHT) [83,84,85]. Within the AGM, HSCs and non-self-renewing progenitors arising from hemogenic endothelium during EHT assemble within intra-aortic hematopoietic clusters that bud predominantly from the endothelial floor of the arterial DA. Progression from HE to HSCs in vivo induces sequential upregulation of the hematopoietic cell surface markers CD41 (in murine models), CD43 and CD45, while expression of endothelial markers, such as CD144 and CD31, is gradually lost during this transition [86,87].

#### 2.4.2. Regulatory Determinants of EHT and HSC Emergence

The transcription factor RUNX1 is the primary regulator of the EHT process [34,53,54,55,88], enabling the progressive loss of endothelial gene expression and concomitant activation of definitive hematopoiesis during development [89,90,91]. RUNX1 expression is upregulated by GATA2 in hemogenic endothelium [92]. In close interaction with its co-factor CBFβ, RUNX1 promotes the hematopoietic program by remodeling the cellular epigenetic landscape, opening previously inaccessible genetic neighborhoods to hematopoiesis-specific transcription factors [93]. Transcriptional repressors GFI1 and GFI1B are direct targets of RUNX1 [94]. They participate in active emerging hematopoiesis by downregulating the endothelial program in HE through recruitment of the coREST:LSD1:HDAC1:HDAC2 epigenetic remodeling complex. Histone deacetylases (i.e., HDAC1 and HDAC2) recruited within the complex repress negative regulators of the TGF-β signaling pathway, suggesting a functional requirement for TFGβ signaling during EHT [88,94,95,96,97,98]. Multiple genes bound by GFI1 and GFI1B also contain NOTCH binding motifs, consistent with a crosstalk between GFI1 repressors and NOTCH pathways for EHT during development [95]. Unexpectedly, the master regulator SCL/Tal1, characteristically known for its role in early onset hematopoiesis (see Section 2.2.3), was also implicated in the regulation of the EHT process in a murine PSC differentiation system [99].

Recent advances have provided additional insights into the molecular mechanisms that govern endothelial-to-hematopoietic transition. Numerous studies have highlighted the importance of sterile inflammatory signals mediated by the cytokines tumor necrosis factor-α (TNF-α) [100], interferon (IFN) types I/II [101,102], interleukin-6 (IL6) [103] and inflammasome-derived interleukin-1β (IL1-β) [104], as well as signaling through the cGAS-STING pathway [105] and downstream of Toll-like receptor 4 (TLR4) [106] and RIG-I-like receptors (RLRs) [107] in promoting definitive hematopoietic development from hemogenic endothelium. In several studies, a NOTCH inflammatory signaling network was identified in the regulation of HSC emergence but the specifics of this cooperation remain uncertain.

The role of hypoxia and metabolic pathways in response to oxygen concentration and nutrient availability (e.g., glucose, glutamine, pyruvate) during EHT has also been underscored in various model systems [36,108,109,110,111]. In particular, progression through the EHT stage involves a glycolysis to oxidative phosphorylation metabolic shift induced by onset of circulation and laminar shear stress within the embryo [36,111]. Hypoxia-inducible factor-1α (HIF-1α), a master regulator of the cellular response to hypoxic signals, integrates nutrient availability and hematopoietic production by sensing reactive oxygen species (ROS) production mediated by fluctuating metabolic demands during embryonic development [108].

The regulatory determinants of EHT and HSC emergence continue to expand in complexity, incorporating signaling axes triggered by the biomechanical forces of blood flow [78,79,80,112,113,114,115], catecholamine-mediated signals from the sympathetic nervous system [116], regulators of N-glycan biosynthesis in endothelial cells [117], vitamin C-dependent epigenetic modulators of chromatin state [118], and the molecular machinery regulating endothelial cell cycle progression [119]. A recent genome-wide single-cell transcriptomic analysis of human hematopoietic tissues during gestation has established a unique molecular identity for arterial VE (IL33, ALDH1A1), HE (ALDH1A1, KCNK17, RUNX1) and nascent human HSCs arising from HE within the AGM (RUNX1, HOXA9, MECOM, MLLT3, HLF, SPINK2), providing unique insights into regulatory determinants of EHT in humans [30]. For instance, the expression of ALDH1A1 and other catabolic enzymes linked to retinoic acid metabolism (e.g., CYP26B1) in arterial VE and HE suggests that RA signaling is required for HSC emergence from these cells. Moreover, the nascent HSC molecular signature discriminates HSCs from progenitors during gestation and implicates distinct regulators of HSC specification (e.g., RUNX1, HOXA9 [120,121]) and self-renewal (e.g., MLLT3 [122], MECOM [123], HLF [124]) for HSCs to arise from precursor HE (Figure 2).

### 2.5. HSC Maturation

Following their emergence within the AGM, nascent HSCs undergo maturation through complex interactions with niche constituents of various hematopoietic sites during development. The functional significance of the AGM region in HSC maturation is supported by numerous studies, but the complexity of the molecular landscape remains to be fully deconvolved [87,125,126,127,128].

Nascent HSCs delaminate from the AGM vascular wall, enter the peripheral circulation and colonize the fetal liver at around 6 weeks of gestation in human embryos, where they undergo considerable expansion and continue to mature in response to microenvironmental cues [30,37,129]. Liver HSCs gradually suppress regulators of the fetal program (e.g., LIN28B, HMGA2) [130,131] and genes linked to endothelial surface proteins (e.g., CD144) and cytokines receptors involved in early HSC development (e.g., IL3RA). They enter a homeostatic state by repressing oxidative phosphorylation, spliceosome and ribosome function, as well as glucose and nucleotide metabolism. Concomitantly, maturing HSCs within the liver progressively upregulate expression of key regulators of HSC self-renewal and engraftment (e.g., MLLT3 [122], HLF [124] and MSI2 [132]), and cell surface proteins, such CD133 (encoded by PROM1), L-selectin (encoded by SELL), CD52, MHC class II (e.g., HLA-DRA) and other genes involved in antigen presentation [30] (Figure 2).

Definitive HSCs subsequently lodge to the spleen [133] and bone marrow [134]. After birth, splenic hematopoiesis is transient and the bone marrow becomes the primary site of hematopoiesis [135]. Although post-natal marrow HSCs share some transcriptional regulators with AGM and liver HSCs (e.g., RUNX1, HOXA9, MECOM, MLLT3, HLF, SPINK2), they adopt a more dormant state with reduced rates of protein synthesis [136], shift to an anaerobic glycolytic metabolism with reduced ROS levels [137,138,139,140], transition from innate to adaptive immune production [141,142,143] and display increased expression of various surface markers (e.g., CD62L, CD52, HLA-DRA) [30] (Figure 2). Recent single-cell chromatin accessibility and transcriptomic analyses applied to post-natal hematopoietic cells [32,33,35,144,145] have further characterized mature cellular populations after their emergence. They have exposed marked heterogeneity within phenotypically defined adult stem and progenitor cell populations and have highlighted a differentiation continuum distinct from the classic hierarchical stepwise model of hematopoiesis (reviewed in [146,147,148,149]). A developmental switch from a fetal liver HSC erythroid-biased output [129,131] to an adult HSC megakaryocytic-primed hematopoiesis [150] was also highlighted in these studies.

## 3. Intrinsic Cues for De Novo Generation of HSCs by Indirect Reprogramming

Currently employed human PSC differentiation approaches have been largely extrapolated from findings in various model organisms. Most protocols utilize serum-free media and the stepwise addition of cytokines and morphogens to recapitulate in vitro the diverse intrinsic cellular signals known to drive hematopoiesis in vivo. This section examines key intrinsic cues modulated in culture for the de novo generation of human HSCs, with a focus on stage-specific cell signaling pathways, transcription factors (TFs) linked to HSC regulation and epigenetic regulators of hematopoiesis (summarized in Table 1).

### 3.1. Cell Signaling Pathways

Although critical for the production of engrafting HSC in vivo, intrinsic signaling pathways manipulated during PSC differentiation ex vivo have thus far exclusively promoted the generation of extra-embryonic-like definitive hematopoiesis. As outlined in this section, further refinement of our understanding of stage-specific cell signaling pathways is required to enable efficient production of long-term repopulating HSCs ex vivo.

#### 3.1.1. Cell Signaling Regulators of Mesodermal Induction and Specification Ex Vivo

Several ligand molecules, namely BMP4 [18,151,152,153,154,155], FGF2 [152,153,155,156], Activin A [152,155,157], WNT3a [154], WNT11 [158], R-spondin2 [159] and VEGFA [151,153,155], and the signaling pathways they trigger upon binding to their target cells can promote selective differentiation of human PSCs to mesodermal progenitors and their specification to hematoendothelial fates ex vivo. Understanding the intracellular signals regulating the earliest stages of hematopoietic development ex vivo has also advanced our knowledge of how the primitive and definitive hematopoietic programs are differentially regulated by distinct signaling pathways [160,161]. In these studies, the addition of a small molecule repressor of the Activin/Nodal pathway (SB431542, SB) at the mesodermal stage of differentiation downregulated the primitive program [160], while activation of the WNT/β-catenin pathway with CHIR99021 (CHIR) enriched the definitive program, as measured by T-lymphoid differentiation potential [161]. Moreover, expression of a single cell surface marker (CD235a+) was sufficient to mark and separate mesoderm fated to a primitive program from CD235a− mesoderm, giving rise to definitive hematopoiesis [161]. On the basis of these findings, most PSC differentiation protocols now include CHIR/SB in the early phase (day 2–3) of differentiation, and CD235a+ mesodermal cells are generally excluded to favor the emergence of the definitive hematopoietic program. However, these manipulations alone are inadequate for the generation of cellular populations capable of long-term engraftment in vivo, implying that other pathways may be aberrantly modulated during ex vivo differentiation of human PSCs.

#### 3.1.2. Cell Signaling Regulators of Endothelial Arterialization Ex Vivo

To promote arterialization of vascular endothelium ex vivo, multiple studies have independently activated key signaling pathways implicated in this process. One approach supplemented the culture medium at the mesodermal phase of human PSC differentiation with LY294002, a small molecule inhibitor of PI3K kinase, for indirect activation of the MAPK/ERK signaling pathway [75]. The formation of HE expressing arterial markers (e.g., DLL4) was robustly enhanced. Consistent with a definitive hematopoietic program, blood cells that arose from arterialized HE cells were highly enriched in lympho-myeloid progenitors and red blood cells expressed high levels of adult β-globins. In contrast, non-arterialized HE populations produced a more primitive-type hematopoiesis.

When NOTCH signaling was manipulated, either by the addition of Resveratrol during mesodermal specification [162], by a 4-day incubation of human PSC-derived immature HE progenitors on immobilized NOTCH ligand DLL1 [163], or by culture under hypoxic conditions [185], arterial-type HE also emerged in culture. DLL1-mediated activation of NOTCH increased definitive lympho-myeloid hematopoiesis from arterial HE compared to untreated controls [163]. Although NOTCH signaling is essential for endothelial arterialization and the subsequent emergence of a definitive hematopoietic program, manipulation of this pathway alone was insufficient to promote the development of engraftable HSCs. Future studies are needed to assess whether a combined approach to modulate Activin/Nodal/TGFβ, WNT/β-catenin, MAPK/ERK and NOTCH signaling pathways might provide a synergy sufficient to further promote definitive hematopoiesis and the generation of engrafting HSCs.

#### 3.1.3. Cell Signaling Regulators of Endothelial-to-Hematopoietic Transition Ex Vivo

Ectopic activation of key signaling pathways linked to the EHT process can be exploited to facilitate the emergence of definitive hematopoiesis in PSC differentiation cultures. The enhanced hematopoietic development observed following DLL1-mediated stimulation of NOTCH signaling in immature HE (see Section 3.1.2) was unambiguously attributed to increased endothelial-to-hematopoietic transition rather than improved survival or proliferation of hematopoietic cells at the post-EHT stage of differentiation [163]. Likewise, engineering a potent NOTCH signaling environment during EHT by the addition of NOTCH ligand DLL4 and adhesion molecule VCAM1 enhanced definitive hematopoiesis, as evidenced by the embryonic-to-fetal globin switch, the increased output of multilineage CFUs, robust T-lymphoid competency and strong transcriptional correspondence with primary HSCs [164].

Modulation of the TGFβ signaling pathway with the small molecule inhibitor SB431542 promotes EHT and increases hematopoietic output from HE during PSC differentiation ex vivo [97,155,165,186,187,188,189]. However, the mode of action of SB431542 and differences in dosage or timing of its addition during in vitro differentiation have resulted in apparent contradictions (reviewed in [190]). In particular, although widely recognized as a potent inhibitor of Activin and TGFβ signaling, SB431542 paradoxically increases phosphorylation of select TGFβ intracellular mediators (i.e., SMAD2 and SMAD3) [97]. Overexpression of constitutively active SMAD2/3 alone recapitulated the effects of SB431542 [97] and the addition of TGFβ1 ligand enhanced hematopoietic output ex vivo [189], suggesting a functional requirement for TGFβ activity during PSC differentiation.

#### 3.1.4. Cell Signaling Regulators of HSC Maturation Ex Vivo

Beyond cell signaling regulators shown to establish a definitive hematopoietic program from PSCs, various combinations of hematopoietic cytokines, namely interleukin-3, interleukin-6, stem cell factor, thrombopoietin and FLT3-ligand, are also required to support hematopoietic commitment and maturation in PSC culture. However, several seminal studies have shown that cytokine- and morphogen-based differentiation protocols fail to provide the signaling cues necessary for complete HSC maturation and robust engraftment ability [18,151,153,191,192,193,194,195,196].

### 3.2. Transcriptional Regulators

A network of TFs plays a pivotal role in cell fate determination during mammalian embryonic development. This understanding has prompted the development of alternative differentiation protocols incorporating ectopic expression of single TFs or combinations of TFs in PSCs or their derivatives. In this section, we discuss how leveraging advances in transcriptional regulation of hematopoietic development could surmount some of the molecular bottlenecks for the generation of functional human HSCs in vitro.

#### 3.2.1. Ectopic Expression of a Single Transcription Factor

Seminal work in the early 2000s [166] set a precedent for this strategy by enforcing the expression of HOXB4, a key homeotic selector gene implicated in the regulation of HSC self-renewal [197,198]. Using an inducible murine ESC system, HOXB4 was overexpressed in EB-derived hemogenic endothelium from day 4 to day 6 of differentiation. Hematopoietic cells that arose in HOXB4-induced cultures displayed long-term engraftment potential in both primary and secondary recipient mice. However, the donor grafts exhibited marked myeloid skewing and lymphoid engraftment waned over time, indicating that overexpression of HOXB4 alone was insufficient to fully convert primitive ESC-derived progenitors into bona fide HSCs [166]. Although several subsequent studies using human PSCs confirmed that overexpression of HOXB4 increased hematopoietic progenitor cell formation in vitro [167,168,169,170], these investigations globally failed to generate human HSCs with engraftable properties in vivo [167,169]. Notwithstanding data from murine studies, expression of HOXB genes in human ESC-derived hematopoietic cells was elevated compared with bona fide umbilical cord blood CD45+CD34+CD38- cells. Instead, genes of the HOXA cluster were downregulated, suggesting that HOXA genes may be a preferred target for ectopic expression to augment definitive hematopoietic potential of human PSCs [167].

In the past decade, there have been additional attempts to optimize de novo generation of human HSCs by enforcing expression of a single hematopoietic TF during PSC differentiation. Illustrative examples include SCL [171,172,173], RUNX1a [175], RUNX1c [176,177], SOX17 [76,174], HOXA9 [178] and MLL-AF4 [179]. Transplantable HSCs were typically not detected in these studies. The protocol based on RUNX1a overexpression gave rise to significant human cell engraftment nine weeks after transplantation [175]. However, ex vivo cultures were largely composed of undifferentiated CD34+ cells, and most transplanted mice did not survive long-term. Because aberrant RUNX1 expression has been linked to abnormal hematopoiesis and hematologic malignancies [199], these observations raised concern that malignant transformation had likely occurred. Similarly, enforced expression of the MLL-AF4 fusion protein in hematopoietic cells differentiated from human PSCs resulted in up to 60% engraftment in primary and secondary immune-deficient mouse models, but B-cell leukemia transformation was demonstrated after transplantation [179]. Consistent with these findings, although MLL has an active role in controlling key regulators of HSCs (e.g., HOX and MEIS1 genes) [200,201], MLL-AF4 is also the product of the t(4:11) chromosomal translocation frequently found in infant B cell acute lymphoblastic leukemia [202]. Globally, these studies pointed to species-specific differences and suggested that overexpression of a single TF is unlikely sufficient to alter in vivo functionality of PSC-derived cells.

#### 3.2.2. Ectopic Expression of Transcription Factor Combinations

Recent investigations have examined the combinatorial use of TFs to enable the formation of HSCs from pluripotent stem cells. To date, at least five combinations of TFs have been reported (Table 1). In one study, ectopic expression of five TFs (ERG, HOXA9, RORA, SOX4 and MYB) endowed PSC-derived hematopoietic progenitors with self-renewal capacity, allowing their expansion ex vivo and repopulation in NSG mice. However, the engraftment was transient and restricted to myeloid and erythroid lineages [180]. Two independent transcriptional combinations also promoted the differentiation of human PSCs towards an endothelial intermediate that subsequently gave rise to hematopoietic cells with restricted pan-myeloid (ETV2 and GATA2) or erythro-megakaryocytic (GATA2 and TAL1) potential [181]. However, meaningful engraftment was not observed in these experiments. Following observations of silencing of most medial HOXA genes in hematopoietic cells differentiated from human PSCs, genes encoding HOXA5, HOXA7 and HOXA9 were overexpressed or induced by retinoic acid during EHT in culture [120]. This strategy enhanced the definitive hematopoietic program ex vivo but was unable to promote multilineage reconstitution after transplantation.

To date, only one study has provided convincing evidence of robust, long-term engraftment potential without malignant transformation using a complex combination of seven transcription factors [182]. In the initial steps of this approach, a population of hemogenic endothelium fated to a definitive hematopoietic program was differentiated from human PSCs and cultured for an additional 3 days under conditions promoting endothelial-to-hematopoietic transition. At the end of the culture period, cells were transduced with a library of 26 TFs independently cloned into doxycycline inducible lentiviral vectors. Transduced cells were immediately transplanted within the bone marrow of adult immune-deficient mice. This step was critical to complete HSC maturation by exposing PSC-derived cells to signals from the marrow niche where adult HSCs normally reside. Remarkably, a fraction of the recipient animals showed clear evidence of multilineage engraftment. Molecular studies pinpointed seven transcription factors (ERG, HOXA5, HOXA9, HOXA10, LCOR, RUNX1, SPI1) that had likely conferred multi-lineage hematopoietic reconstitution potential. HSCs generated by induction of these seven TFs alone could engraft when transplanted into recipient mice, giving rise to all major blood lineages long-term. Importantly, engrafted cells could reconstitute multilineage hematopoiesis in secondary recipients, providing evidence of self-renewal properties.

Although these results represent a major step forward, there are some notable limitations. The reliance on an in vivo maturation step, the oncogenic potential of most TFs and the modest differentiation efficiency preclude clinical translation of this approach. Moreover, transcriptional analysis of hematopoietic cells differentiated from PSCs indicated a strong correlation with an HE signature. Hence, further studies are needed to more fully recapitulate human HSC maturation in culture. Recent investigations have compared at single-cell resolution the transcriptome profile of in vitro-generated HSC-like cells with bona fide HSCs within fetal liver [30,37], cord blood [30] or peripheral blood sources [35]. The integrated data identified key transcriptional regulators with limited expression in PSC-derived cells relative to their in vivo counterparts. Systems biology algorithms, such as CellNet [203,204], CellRouter [205], FateID [206], SingleCellNet [207] and CellComm [208], have been critical to integrate transcriptomic and other molecular parameters to comprehensively understand the complex regulatory network that govern the emergence of HSCs in vivo. Methods for de novo production of HSCs will undoubtedly benefit from these new insights in coming years.

### 3.3. Epigenetic Regulators

Although efforts to produce HSCs ex vivo have largely focused on re-activation of critical regulators that promote HSC differentiation and maturation, emerging evidence indicates that the acquisition of HSC multipotency and engraftment potential during embryonic development is restricted by a distinct epigenetic barrier [183,184]. In particular, the polycomb repressor complex 2 (PRC2) participates in transcriptional repression during hematopoietic development via methylation of lysine residue 27 of histone 3 (H3K27me3) at target genes [209,210]. PRC2 consists of multiple core subunits, including EZH2 (enhancer of zeste homolog 2) and its homolog EZH1 [209,211,212]. Remarkably, knockdown of EZH1, but not EZH2, enhanced the formation of functional definitive HSCs in embryonic murine and zebrafish models in vivo and promoted multi-lymphoid output from human PSCs in vitro [183,184]. Thus, inhibition of key epigenetic regulators in combination with synthetic activation of HSC transcription factors may be required to overcome the protracted cellular maturation in PSC differentiation systems and promote the full functionality of HSCs engineered ex vivo. Proof-of-concept of this approach was recently demonstrated in a PSC differentiation model of neural progenitor cells, whereby pharmacologic inhibition of the epigenetic factors EZH2, EHMT1/2 or DOT1L at early stages of differentiation enabled rapid molecular and physiological maturation of newly born neurons in culture [213].

## 4. Extrinsic Cues for De Novo Generation of HSCs by Indirect Reprogramming

Intrinsic factors alone are unlikely sufficient for de novo generation of engraftable HSCs from pluripotent stem cells. During development, hematopoietic cells emerge and mature within three-dimensional specialized microenvironments or niches within the AGM, liver, bone marrow and other anatomical sites. The array of environmental signals guiding hematopoietic development in vivo was recently reviewed [214]. Recapitulating this complexity in vitro is challenging but deemed important to overcome bottlenecks in HSC specification. This section illustrates how extrinsic environmental cues, namely cell–cell contact, cell–matrix interactions, diffusible factors and biomechanical forces, may be integrated within human PSC differentiation protocols to enhance the emergence of definitive hematopoiesis (summarized in Table 2).

### 4.1. Cell–Cell Contact

Methods for manufacturing hematopoietic stem and progenitor cells from PSCs often rely on coculture with monolayers of cells derived from distinct anatomic tissues to mimic the cellular signals arising from developmental niches in vivo. Characteristic examples of this approach are summarized in Table 2.

Coculture with murine marrow stromal cell lines or fetal liver- and AGM-derived stromal cells during differentiation markedly enhanced the formation and maintenance of hematopoietic progenitors. In one report [223], although PSC-derived hematopoietic cells displayed engraftment capacity in primary and secondary murine models, the lack of adult globin expression and limited lymphoid contribution to the graft implied incomplete HSC specification. Moreover, the use of undefined animal-derived sera in these studies muddles signaling interactions driving differentiation, is prone to lot-to-lot variation and limits clinical translation.

Recent studies have exposed the role of macrophages as microenvironmental cellular regulators in hematopoietic development [224,236,237,238,239,240,241,242]. In vertebrate models, macrophages supply pro-inflammatory signals that support the emergence, expansion and maturation of HSCs [236,237,238]. They also enhance nascent HSCs and hematopoietic progenitor cell mobilization at the embryonic aorta wall [240,242], influence their homing and retention within vascular niches [241] and determine hematopoietic clonality by patrolling and removing populations of stressed HSCs and hematopoietic progenitor cells during development [239]. Although macrophages are important cellular components of the HSC inductive niche, their impact on PSC differentiation ex vivo remains largely unexplored. In a single study, ex vivo coculture of human macrophages alone or in combination with marrow stromal cells enhanced the formation of human HSCs and hematopoietic progenitor cells from pluripotent stem cells. The macrophage–stromal cell combination, in particular, enabled the production of cells with phenotypic characteristics of human HSCs but limited long-term engraftment in fetal sheep after in utero transplantation [224].

Recapitulating the vascular niche with endothelial cells during PSC differentiation had minimal impact on hematopoietic yield. However, coculture with monolayers of endothelial cells engineered to express membrane-bound JAG1 and DLL4 NOTCH ligands upregulated RUNX1 and GATA2 expression and promoted definitive hematopoiesis. Remarkably, multilineage and long-term reconstitution of the hemato-lymphopoietic system was observed in recipient mice. Although this study marked a noteworthy technical advance, co-infusion of engineered endothelial cells was required to enable homing and complete maturation of transplanted cells in vivo, thereby restricting its clinical pertinence [225].

The importance of cellular microenvironmental cues for de novo generation of HSCs was also demonstrated in PSC differentiation strategies based on teratoma formation [226,227,228,229]. In this two-pronged approach, teratoma-bearing immunodeficient mice were first produced by subcutaneous or intratesticular injection of PSCs, and HSCs and hematopoietic progenitor cells that arose over time within the permissive teratoma niche were then isolated and assayed for engraftment potential by transplantation into recipient mice. The teratoma formation step generally required co-injection of OP9 stromal cells [226,227], exogenous administration of hematopoietic cytokines [227], enforced expression of combinations of TFs [228] or alternative host genetic backgrounds [228,229] to fully support blood cell development in vivo. Upon transplantation, teratoma-derived primitive hematopoietic cells displayed long-term trilineage reconstituting activity without leukemic transformation. Although recapitulation of functional hematopoiesis within teratoma tumors has no clinical bearing, this approach has provided direct evidence that adequate cellular signals from the niche enables PSC differentiation to populations of HSCs with functional integrity in vivo.

### 4.2. Cell–Matrix Contact

Within anatomically defined niches, PSCs and their differentiated progeny receive and integrate regulatory signals from extracellular matrices (ECM) during development. To bypass constraints of feeder cells and achieve more defined ex vivo culture conditions, natural and synthetic matrices are commonly utilized for the maintenance and expansion of human pluripotent stem cells. Crude ECM extracts derived from murine sarcoma or xeno-free human tissues, such as Matrigel and Geltrex, as well as native or recombinant laminin, collagen, vitronectin, tenascin or fibronectin, key protein components of the ECM, provide a dependable matrix support in PSC culture systems [243,244].

Although most studies rely on Matrigel coating for hematopoietic differentiation of PSCs [230], other ECM constituents display hematopoiesis-inducing potential in culture. In one study, vitronectin-coated vessels were shown to augment early hematopoietic development relative to customary Matrigel support by promoting blood-fated mesoderm and HE generation from mesodermal progenitors. This outcome was mediated by intracellular signaling induced upon binding of vitronectin to integrin receptors αvβ3 and αvβ5 in mesodermal cells. Differences in hematopoietic development between Matrigel and vitronectin matrix supports were imputed in part to inhibitory constituents found in Matrigel crude extracts compared to highly purified vitronectin preparation [231].

Another report [232] stemmed from the observation that OP9 stromal cells are superior to MS5 and S17 coculture systems to support hematopoiesis [193]. By directly comparing the transcriptomic landscape of each cell line, it was shown that tenascin C was uniquely expressed in OP9 feeder cells and could provide potent cell–matrix contact for hematoendothelial differentiation of pluripotent stem cells. This finding cohered with the known function of Tenascin C in embryonic and post-natal hematopoiesis [245,246,247]. Tenascin C enabled efficient generation of definitive multilineage hematopoietic progenitor cells, but these cells lacked engraftment potential in murine models. Hence, although components of the extracellular matrix offer a simple, chemically defined platform for the generation of clinical grade blood cells, adjunct maturation signals will be required to activate the self-renewal program in hematopoietic progenitors generated on ECM supports.

### 4.3. Soluble Factors

Hematopoietic stem cell formation during development is closely regulated by a number of soluble signaling molecules secreted systemically or locally by cellular constituents of microenvironmental niches [214]. Recent investigations have integrated soluble factors in standard protocols for the hematopoietic differentiation of human PSCs ex vivo. An illustrative example is the large family of pro-inflammatory cytokines. In one study, a physiological burst of IL-1β was provided by transient inflammasome activation in human PSC differentiation cultures poised to undergo EHT [104]. A more robust multilineage CFU production and increased emergence of CD34+CD45+ phenotypic hematopoietic progenitors expressing RUNX1c suggested improved hematopoietic commitment. However, no selective advantage was observed in cells with multilineage engraftment potential, as evidenced by the overall lack of human CD45+ cell populations in transplanted mice. Moreover, exogenous supplementation of inflammatory cytokines (i.e., TNFα, IFNɣ and IL-1β) did not improve hematopoietic differentiation of human PSCs [233], suggesting that optimized timing and duration of inflammatory signals in culture are required.

In recent years, improved genomics technologies have provided more comprehensive insights into spatial and cell-type-specific gene expression, and enabled the identification of novel factors and extrinsic ligand–receptor pairs during hematopoietic development. One study employed laser capture microdissection coupled with RNA sequencing to map ventrally polarized genes and signaling pathways associated with HSC emergence [38]. Genomic data generated from this approach identified pathways not previously linked to HSC development. The cardiac epidermal growth factor pathway and its primary regulator, endothelin-1 (EDN-1), displayed marked ventral enrichment and were further assessed for their potential role in the hematopoietic niche. Endothelin-1, an endogenous secreted peptide, was most abundantly produced from endothelial cells, and inference from patterns of endothelin receptor expression in various cellular populations suggested a direct effect of EDN-1 in the process of endothelial-to-hematopoietic transition. In EB-based murine and human PSC differentiation approaches, the addition of EDN-1 during EHT significantly enhanced the formation of early multipotential hematopoietic progenitors (CFU-GEMM) in culture. Improved long-term repopulation was observed after transplantation of murine PSC-derived cells, but the engraftment potential of human HSCs and hematopoietic progenitor cells differentiated ex vivo in the presence of EDN-1 was not investigated. Additional ventrally polarized secreted factors were uncovered in that study and in parallel work employing genome-wide RNA tomography sequencing on the AGM from stage-matched human, chick, mouse and zebrafish embryos [39]. These advances illuminate novel aspects of developmental hematopoiesis, but further work is necessary to assess their relevance for de novo generation of human HSCs.

A growing list of soluble regulators have been implicated in HSCs and hematopoietic progenitor cell production during development (reviewed in [214]), such as hormonal messengers (e.g., estrogen [248], thyroid hormone [249], glucocorticoids [250], cannabinoids [251], and vitamin D [252,253]), as well as cellular metabolites (e.g., glucose [108] and retinoic acid [120,234,235]). In human PSC differentiation systems, the activation of retinoic acid signaling during EHT induced the definitive hematopoietic program but was insufficient to impart full self-renewal capacity in human HSCs [120,234,235]. Mechanistically, exogenous supplementation of retinoic acid to PSC-derived endothelial cells enhanced medial HOXA gene expression by modulating chromatin accessibility [120]. Moreover, mesodermal expression of CDX4, a caudal-like homeobox transcription factor regulated in part by the retinoic acid pathway [254], was shown to promote definitive hematopoietic specification from human pluripotent stem cells ex vivo [255]. However, retinoic acid effects vary with developmental stages, and a high dosage can be detrimental for human HSC development and maintenance [235].

### 4.4. Biomechanical Forces

An array of biomechanical inputs contribute to HSC development in the embryo (reviewed in [256,257,258]). In particular, hemodynamic forces induced by blood flow within the dorsal aorta of developing embryos have emerged as critical regulators of hematopoiesis in vivo [78,79,80,113,114,259,260]. Indeed, a marked reduction in HSC populations was observed in mutant zebrafish and murine models lacking a heartbeat and blood circulation [78,79]. The pulsating aorta generates distinct biomechanical forces that influence vascular and hematopoietic cell formation during development, including frictional wall shear stress (WSS), circumferential stretch (CS) and radial hydrostatic pressure. Although the mechanisms by which hemodynamic forces are sensed and transduced within vascular and hematopoietic cells during ontogeny remain poorly understood, landmark studies have provided novel insights into the complexity of these interactions. Notably, frictional WSS forces were shown to induce nitric oxide signaling in vivo [78,261] and trigger an influx of cytosolic calcium, leading to upregulated production of prostaglandin E2, a key lipid mediator implicated in the regulation of HSC homeostasis in vertebrates [114,262]. In turn, prostaglandin E2 influenced specification and generation of definitive HSCs from hemogenic endothelium by activating the cyclic AMP-protein kinase A-CREB signaling axis and the downstream WNT and NOTCH developmental pathways [114].

How biomechanical forces can be leveraged for de novo generation of human HSCs will require further investigation. One recent proof-of-concept study used human iPSC-derived HE cells to engineer a novel microfluidic organ-on-a-chip model of the human dorsal aorta and address the impact of flow-related forces on HSC and hematopoietic progenitor cell formation ex vivo [115]. Enhanced hematopoietic induction was observed under culture conditions that emulate WSS or CS compared to a conventional static environment. Distinct from the molecular mechanisms underpinning WSS biomechanical cues, CS-mediated forces activated intracellular signaling through YAP (Yes-activated protein), an established transcriptional regulator of cell proliferation and mediator of mechanical stimuli, resulting in increased expression of RUNX1 and production of multipotent hematopoietic colony formation in CFU assay. YAP was essential to maintain the hematopoietic program but dispensable for the specification of HE within the aorta and initiation of RUNX1 expression. Importantly, activation of YAP was mediated by the Rho family of GTPases, providing a molecular link between CS hemodynamic forces and YAP activation during HSC and hematopoietic progenitor cell formation. Although recapitulating biomechanical forces for large-scale therapeutic production of HSCs and hematopoietic progenitor cells ex vivo is cumbersome, pharmacologic activation of YAP via Rho-GTPase stimulation would offer a more practical approach for de novo generation of HSCs in vitro. To test this possibility, Rho GTPases were transiently activated during EHT in a standard iPSC differentiation culture system with pharmacological compounds CN03/CN04. Treatment with CN03 or CN04 significantly enhanced RUNX1 expression and promoted hematopoietic specification ex vivo, as measured by flow cytometry and clonogenic assays. However, future studies will be necessary to assess the impact of YAP stimulation on the long-term repopulating potential of HSCs generated ex vivo using this approach.

## 5. Conclusions and Future Perspectives

The ability to produce human long-term engrafting HSCs in a dish has far-reaching implications in clinical medicine. Studies in animal and cellular model systems have provided a foundational understanding of hematopoietic ontogeny and, in the past decade, the genomics revolution has delivered new transformative insights that set the stage for imminent breakthroughs in this field. In particular, the recent creation of a single-cell transcriptomic map of human HSC ontogeny now enables the direct comparison of transcriptional similarity between PSC-derived HSCs and distinct native cell types at various stages of hematopoietic maturation. A close transcriptional match between HSCs engineered ex vivo and native immature AGM or extraembryonic HSCs has confirmed their protracted progression to more mature liver and marrow stages in current ex vivo PSC differentiation protocols [30]. Thus, further understanding of cell-intrinsic/extrinsic signal combinations that drive human HSC maturation, and the elaboration of novel in vitro platforms that recapitulate in vivo maturation processes with increased spatial and temporal fidelity (e.g., organoids, gastruloids and organ-on-a-chip), will be critical to access the full potential of PSC technologies in modeling and treating disorders of adult human HSCs.

Inadequate activation of core transcription factor networks defining HSC identity could explain, in part, the incomplete maturation and lack of engraftment potential in HSCs generated ex vivo. In previous studies, HSC-specifying TFs utilized to drive PSC differentiation were selected based on evolutionary conservation or enrichment in HSCs compared to more mature progenitors in published gene expression datasets. Recent investigations suggest that genes controlling unique cellular identities are driven by so-called “super-enhancers (SEs)” [263,264]. Importantly, SEs are bountiful in cell-type-specific TF binding motifs that enable TFs to bind cooperatively. Novel genomics approaches have identified putative SEs in more than 100 human tissue or cell types, including hematopoietic progenitors [264]. However, their identification is lacking in highly enriched populations of HSCs. Functional characterization of HSC-specific SEs could help establish a more comprehensive list of master TFs defining HSC identity (Li et al., manuscript in preparation). Ectopic activation of these master regulators, alone or in combination with inhibition of key epigenetic regulators, during PSC differentiation could improve de novo generation of engraftable human HSCs.

Efficient homing, retention and survival within specialized niches of the recipient marrow immediately after transplantation constitute a sine qua non for long-term engraftment of HSCs produced in vitro. An early transplantation failure of PSC-derived HSCs and hematopoietic progenitor cells was uncovered after direct intra-femoral injection in xenograft murine models [265]. The inability of these cells to home and survive after transplantation was attributed to limited expression of the CXCR4 chemokine receptor, a well-known mediator of hematopoietic stem and progenitor cell marrow homing and egress in vivo [266,267,268]. Ectopic expression of CXCR4 restored ligand (CXCL12)-mediated signaling and improved short-term marrow retention. However, CXCR4 alone was insufficient to confer sustained engraftment. Future studies are necessary to optimize the timing and dosage of CXCR4 expression during differentiation and target alternative surface molecules implicated in cellular adhesion (e.g., integrins, cadherins, selectins, tetraspanins) to generate human HSCs with short-term retention capability and long-term engraftment competency in vivo.

Much work is still required to confirm the potential of these new avenues to comprehensively integrate intrinsic cell signaling and extracellular cues to produce clinical-grade adult-like HSCs. The prospect of this approach to provide durable benefits to human health, however, justifies continued optimism and increasing efforts toward novel scientific advances and optimized ex vivo differentiation schemata of human pluripotent stem cells.

## Figures and Tables

**Table 1 cells-12-00321-t001:** Intrinsic cues for de novo generation of HSCs from human PSCs.

Intrinsic Signals	Regulator (Pathway)	References
Cell signaling pathways	BMP4 (BMP) FGF2 (FGF) Activin A (Nodal) WNT3a (canonical WNT) WNT11 (non-canonical WNT) R-spondin2 (TGFβ) VEGFA (VEGF) SB431542 (Nodal)	[18,151,152,153,154,155] [152,153,155,156] [152,155,157] [154] [158] [159] [151,153,155] [160]
	CHIR99021 (canonical WNT)	[161]
	LY294002 (MAPK/ERK) Resveratrol (NOTCH) DLL1 (NOTCH)	[75] [162] [163]
	DLL4 (NOTCH) SB431542 (TGFβ)	[164] [165]
Transcriptional regulators	HOXB4	[166,167,168,169,170]
	SCL	[171,172,173]
	SOX17	[76,174]
	RUNX1a	[175]
	RUNX1c	[176,177]
	HOXA9	[178]
	MLL-AF4	[179]
	ERG/HOXA9/RORA/SOX4/MYB	[180]
	GATA2/ETV2	[181]
	GATA2/SCL	[181]
	HOXA5/HOXA7/HOXA9	[120]
Epigenetic regulators	ERG/HOXA5/HOXA9/ HOXA10/LCOR/RUNX1/SPI1 EZH1	[182] [183,184]

**Table 2 cells-12-00321-t002:** Extrinsic cues for de novo generation of HSCs from human PSCs.

Extrinsic Cues	Description	References
Cell–cell contact	Murine marrow stromal cell line OP9 Murine marrow stromal cell line S17 Murine marrow stromal cell line C166	[193,215] [216,217,218,219] [216]
	Murine marrow stromal cell line MS5	[220,221]
	Murine marrow stromal cell line AM20.1B4	[222]
	Murine marrow stromal cell line C3H10T1/2 Human and murine fetal liver-derived cells	[15] [218,223]
	Murine AGM stromal cells Human marrow stromal cells Human monocyte-differentiated macrophages Endothelial cells originating from mouse YS Endothelial cells originating from human UCB Cell–cell contact within teratomas	[223] [224] [224] [216] [225] [226,227,228,229]
Cell–matrix contact	Matrigel	[230]
	Vitronectin	[231]
	Tenascin C	[232]
Soluble factors	Proinflammatory cytokines	[104,233]
	Endothelin-1	[38]
	Retinoic acid	[120,234,235]
Biomechanical forces	Blood flow circumferential stretch	[115]

## Data Availability

No new data were created or analyzed in this study. Data sharing is not applicable to this article.

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
