# Peer review of "De Novo Generation of Human Hematopoietic Stem Cells from Pluripotent Stem Cells for Cellular Therapy"

_cells, 2023, doi:10.3390/cells12020321_

Round 1
Reviewer 1 Report
The report provides a comprehensive review on hematopoietic differentiation from pluripotent stem cells. While overall it is a well-written report, minor changes may further improve the manuscript.
Section 1:
Line 62: Pluripotent state is not “transitional” in Yamanaka’s method.
Line 66: “de-differentiation” is better not to be used for iPS generation.
Line 86: Is “2-dimensional culture system” a monolayer cell culture?
Line 115-116: I agree with the authors on non-reproducibility of the work mentioned. Hence the work should not be cited and the sentence “Notably, human multilineage hematopoietic progenitors were produced by enforced expression of the homeodomain transcription factor Oct4, although 116 this observation has not been reproduced [26]. ” can be removed from the manuscript.
Section 2:
From the text it may be implied that EHT is specific to AGM hematopoiesis, while the definitive wave of yolk sac hematopoiesis is also through EHT.
Section 3:
In this section hematopoietic differentiation from PSCs is described in the context of primitive vs definitive differentiation, while as authors mentioned in section 2, HSCs are derived from a third (or forth) wave of hematopoiesis taking place intra-embryonically. The cell intrinsic cues for HSC generation should be sought through stepwise generation of lateral plate like mesoderm and intra-embryonic HE. All the conditions mentioned in section 3.1 can only generate extra-embryonic like definitive hematopoiesis, hence failing to generate repopulating HSC.
Section 4:
All good.
Section 5:
The epigenetic regulators of hematopoiesis could have been described in Section 3, within the cell intrinsic capacity.
Reviewer 2 Report
Jianyi Ding and co-authors Yongqin Li and Andre Larochelle present a strongly written, well-balanced, and very important review of the current status of hematopoietic stem cell generation from pluripotent stem cells in humans. The authors clearly present the necessary background and historical information in this field which is balanced with a review summary of information obtained from model systems. They clearly present the advances and remaining questions/issues in generating HSCs from pluripotent cells in the context of transplantable cells (which are the true test of real HSCs and necessary for the long-awaited promise of these systems). This review is important, well-written, thoroughly cited, and thoughtfully presented. I have no major comments and enjoyed reading this manuscript.
